# Prominent Roles and Conflicted Attitudes of Eumelanin in the Living World

**DOI:** 10.3390/ijms24097783

**Published:** 2023-04-24

**Authors:** Maria Letizia Terranova

**Affiliations:** Dipartimento Scienze e Tecnologie Chimiche, Università degli Studi di Roma “Tor Vergata”, Via della Ricerca Scientifica, 00133 Roma, Italy; terranov@uniroma2.it or mleterranova@gmail.com

**Keywords:** eumelanin, stress responses, radioprotection, radiotropism, anti/pro-oxidant behaviour, fossil record, melanomas, electron transfer, metal ion interactions

## Abstract

Eumelanin, a macromolecule widespread in all the living world and long appreciated for its protective action against harmful UV radiation, is considered the beneficial component of the melanin family (ευ means good in ancient Greek). This initially limited picture has been rather recently extended and now includes a variety of key functions performed by eumelanin in order to support life also under extreme conditions. A lot of still unexplained aspects characterize this molecule that, in an evolutionary context, survived natural selection. This paper aims to emphasize the unique characteristics and the consequent unusual behaviors of a molecule that still holds the main chemical/physical features detected in fossils dating to the late Carboniferous. In this context, attention is drawn to the duality of roles played by eumelanin, which occasionally reverses its functional processes, switching from an anti-oxidant to a pro-oxidant behavior and implementing therefore harmful effects.

## 1. Introduction

No other natural polymer is characterized by so many intriguing properties and is able to fulfill such a wide variety of different functions as eumelanin (called henceforth EU), which is the dark-brown/black component of the melanin family. A lot of unique characteristics make this biopolymer very different from almost every other macromolecule, including the orange-red pheomelanin, the other form of natural melanin.

The first peculiarity that emerges when looking at the natural EU is its not well-defined basic structure, which consists of substructures containing a variable number of 5,6-dihydroxyindole (DHI) and of its 5,6-dihydroxyindole 2-carboxylic acid form (DHICA) units, cross-linked through chemical bonds or physical interactions [1]. A scheme of the synthesis process of DHI and DHICA from the enzyme tyrosinase is shown in Figure 1.

The EU building blocks can be in various oxidation states, giving rise to hydroxyquinone, semiquinone and indoquinone moieties, which are organized prevalently in planar tetramer units. The 3D structure of the biopolymer can also contain an inner porphyrin ring [2,3]. The heterogeneous chemical and structural features of this dark insoluble polymer have been the subject of a series of hypotheses, conjectures and assumptions. The variety of molecular arrangements of natural EU made it indeed hard in the past to obtain a precise description and to identify the characteristics that only quite recently the combined use of advanced analysis methodologies enabled revealing [4,5,6,7,8,9]. Due to its heterogeneity and the lack of long-range order, EU is in general considered an amorphous material, even if diffraction experiments have evidenced a p–p stacking of ≈15 Å sized sheets at interplanar distances of ≈3.7 Å [10].

Both scientific and application-oriented interests led in the last years to a large number of research studies that aimed to elucidate the inherent chemical/physical heterogeneity of EU. It was indeed realized that the “geometrical disorder” derived by the variability of the ratio of DHI to DHICA units, of their oxidation states, of their mutual arrangement and spatial packing, triggers in turn the so-called “electronic disorder” and governs the EU’s distinctive chemical/physical features.

These include the wide electronic delocalization coming from the extended systems of energy bands [11], the switching between redox states [12] and the complex free radical system characterized by the presence of two different paramagnetic centers [13]. Of these, the first is the “intrinsic” one centered on carbon and ascribed to stationary radicals, and the second is the “extrinsic” one, which is related to semiquinone free radicals reversibly generated by irradiations or hydration [12,13]. However, the experimental evidence of these two different radicals presently does not rule out the existence of further radical populations [14].

This combination of rather unusual characteristics accounts for the very specific functional properties of EU. We speak here of the broadband optical absorption coupled with an unusual extinction coefficient [15,16,17], of the radical quenching achieved by reversible redox processes [18], of the effective metal–ion chelation [19,20,21], of the uncommon double electronic/protonic conduction [22] and, last but not least, of the conversion of ionizing radiations [23].

Given the outstanding performances of the naturally occurring pigment, it is not surprising that in the last few years, a variety of EU and EU-based structures have been proposed and tested as multifunctional bio-inspired materials able to play a variety of roles in advanced technologies, from biomedicine to optoelectronic and energy [24,25,26,27,28,29,30,31]. Whereas the material researchers, trying to emulate some of nature’s tricks, attempt to produce synthetic eumelanin exploiting a wide range of possible molecular configurations [32], other researchers are still studying how, why (and when) nature has opted to produce a versatile, multitasking macromolecule through a rational modulation of its chemical/compositional/structural features.

Earth became a biological planet more than 3.5 Ga ago, and life has evolved according to a process of self-organization of chemicals into even more complex networks [33]. Life has been defined by NASA as ‘‘a self-sustaining chemical system capable of Darwinian evolution’’ [34]. However, the occurrence of almost unmodified EU in the present living species looks like this macromolecule has not been touched by a substantial evolution. This odd feature is likely related to the ubiquity in the living word, the extreme preservation, and the conflicting behavior: a combination of peculiarities that make EU unique on Earth.

The aim of the present paper is to look at eumelanin connecting its many, sometimes still unexplained aspects and the various, sometimes contrasting, roles played by this fascinating macromolecule.

## 2. The Ubiquity in the Living World

EU is the unique biopolymer widespread in all the biological world, from vegetables to bacteria, from insects to the more complex animal kingdom, from invertebrates to vertebrates. It is found in fungi, cephalopods, birds, and mammals, just to name a few. In the more complex organisms, EU is present in various organs and plays a variety of roles. As an example, in mammals, EU and its derivatives are found in “substantia nigra”, “locus coeruleus” and “medulla oblongata”, areas of the brain where a series of fundamental neurological functions occur [35]. Here, EU plays a role in neurotransmission and regulates the accumulation of toxic metal ions [36]. The EU pigment is found also located in the retina of the eye and in the cochlea of the ear, where it governs the transmission of functional signals.

The chemico/physical analytical approaches developed in the last two decades have enabled unambiguously identifying the presence of EU also in ancient species and to compare the composition of pigments preserved in the fossil record with those extracted from the latest living organisms.

## 3. The Keeping over Times of Chemical/Physical Features

The impressive preservation of EU over geological time scales is a further interesting feature of this pigment. The highly efficient microbial decomposition does not allow in general the preservation of soft tissue in the fossil record [37,38]. The organic constituents of soft tissues are indeed subjected to diagenetic alterations, i.e., biological and physicochemical degradation processes that cause their transformation into hydrocarbon chains [39], and they are rarely preserved in fossils more than 65 Ma old [40]. However EU has been detected in the ink sacs of coleoid cephalopods dating back to Early Jurassic [41], in fossils about 200 Ma old [42] and also in samples dating to the late Carboniferous, ~307 Ma ago [43]. Appropriate technologies have recently enabled to analyze the physical/chemical characteristics of the fossilized pigments and to confirm the exceptional resilience to diagenesis and the long-term preservation of EU, which are characteristics that distinguish this complex biopolymer from every other biomolecule.

The animal EU is generated by melanocytes, cells able to synthesize the enzyme tyrosinase, and it is contained almost exclusively in lysosome-derived vesicles with a diameter of about 500 nm, which are called melanosomes [44]. Melanosomes are typically found in integumentary structures (feather, scales, skin, hair) of various amniote lineages, from sauropsids (reptiles, birds, non-avian dinosaurs) to synapsids (including mammals) [45,46,47] and distributed in different body tissues and organs of vertebrates and invertebrates [43,48].

The size and shape of melanosomes detected in the fossilized invertebrates are similar to those present in the extant ones, as confirmed by studies performed on coleoid Sepia Officinalis [49]. As regards the vertebrates, fossilized melanosomes have been found also in the retina of the eye, in the inner ear, in some regions of the brain and in other organs [35,50]. A correlation between the external (integumentary) and internal (organs) amount of EU present in living species has been outlined in [51].

The large amount of data obtained from a variety of fossil vertebrates from Carboniferous to Pliocene enables nowadays to hypothesize the possible functions exerted by EU in ancient organisms [46,52].

A deep review reporting and discussing what is presently known about the localization of melanosomes and possible functions in major vertebrate classes has been recently published by Mac Namara et al. [53]. The scheme reported in Figure 2 evidences how in both fossilized and still extant vertebrates, the melanosomes are localized exactly in the same tissues and organs.

It has also been speculated that the history of EU could be backdated up to the prebiotic era [54]. Structural features and physicochemical properties of the melanin found in some living organisms, such as bacteria and fungi, show indeed several analogies with those of the insoluble organic matter found in carbonaceous chondrites, dating 2–4 Ma from the Solar System formation [54].

## 4. The Ability to Withstand Extreme Conditions

The ability of living organisms to synthesize eumelanin imparts a selective advantage in surviving and growing under extreme environments. A striking example is given by the melanized black fungi, which withstand extreme chemical/physical conditions and resist multiple stresses [55]. The outstanding ability of EU to resist the alterations has been tentatively ascribed to the highly cross-linked original state of the biopolymer [39,40]. However, other properties specific to this intriguing macromolecule, such as the complex radical system and the peculiar redox responses, also account for the superior resistance to stresses.

There are several strategies conceived by EU to overcome stresses and to increase the viability of species either living in harsh environments or anyway needing to counteract adverse effects. Stresses include, beyond oxidant agents, extreme temperatures, freezing/defrosting cycles, dryness, changes in osmotic pressure, microbicidal drugs, and the immune response of host organisms. To increase their tolerance to any negative impact, the melanized species react by putting in place the same front-line strategy, namely the boosting of EU production [56].

Examples are given by the human pathogenic *Cryptococcus neoformans*, *Wangiella Dermatitidis*, *Sporothrix schenckii*, *Aspergillus fumigatus*, *Histoplasma capsulatum* and by some phyto-pathogenic fungi, against which the immune systems of the organism under attack implement a defense strategy based on the rapid production of reactive oxygen species (ROS). To counteract the oxidative burst and to maintain/increment the virulence in the host organism, the pathogens increase the EU production. In such a way, at the site of attack, the EU’s efficient system of free radical quenching is able to activate a strong reaction [55,57]. The protective effects of EU against oxidative stress extend to all kinds of reactive species, from singlet oxygen [58] to radicals associated to the excited states of dye molecules [59] and also to those generated by non-photic stimulations [60].

In addition, the great ability of EU to bind redox-active metal ions and oxides is implemented by a variety of organisms, from bacteria to mammals, that utilize the metal uptake process to contrast cytotoxic effects [61].

Due to the effective interaction with metal ions, this biopolymer can exert also a protective function against lipid peroxidation [60]. This further role of EU was discovered when in vitro studies of pigmented and unpigmented liver tissues evidenced the ability of EU to inhibit lipid peroxidation induced by Fe^2+^ ions [62] and also the peroxidation of cardiolipin liposomes induced by Fe^2+^-ascorbic acid [63]. A very complete review of the effects produced in living organisms by the EU–metals interactions can be found in [21].

The stress-induced increase in EU production results also in a decrease in cell permeability, which was an additional defense mechanism for melanized species living in challenging chemical environments. The greater amount of melanin granules increases the cross-linking of the cell walls, significantly reducing the pore sizes and therefore the cell permeability [64,65]. This feature is particularly exploited by melanized species that survive and flourish also under hypersaline conditions [66]. In this case, the increased amounts of EU accumulated in the cells implement an osmo-adaptative strategy that reduces the loss of protective substances from the cell walls [55].

However, the most dangerous among the various stress factors to deal with when survival is at stake is the exposition to high doses of ionizing radiations. In this context, EU shows a further unique feature, i.e., the capability not only to withstand highly ionizing radiations but also to take advantage of them. This functionality distinguishes EU from every other biomolecule, including the also long-surviving chitin and collagen.

The unexpected growth of black fungal colonies exposed to high-level of ionizing radiations observed in Nevada nuclear test sites [67], in areas contaminated by nuclear fallout [68] and in nuclear reactor waters [69], has been fully understood in the 1990s, when Russian researchers deeply investigated the behavior of *Cladosporium sphaerospermum* fungi growing in the highly radioactive areas surrounding the damaged Chernobyl Atomic Energy Station [70,71].

The analysis of the pigmented fungal species thriving in either natural highly radioactive environments or in environments contaminated by anthropogenically originating radionuclides disclosed that, in all cases, such species had a high content of EU. A proof that EU was able not only to assure the viability but also to increase the metabolic activity of living species containing DHI and DHICA oligomers was obtained later by comparing EPR spectra of melanized and non-melanized species exposed to controlled doses of β and γ radiations [23,72].

The significative features detected in the EPR spectra of irradiated cells prompted researchers to investigate more deeply the radiation-induced changes of the EU electronic structure and to study the effects of electron transfer and paramagnetic properties modifications on the viability of melanized species. The EPR analysis performed on the *C. neoformans* cells outlined the occurrence of a stable free radical population with two distinct paramagnetic centers [13], evidencing an unusual complex radical system that has been more recently verified in both natural and synthetic EU [14,73].

Experiments performed on various synthetic EU had demonstrated that even high doses (300 Gy) of γ-rays produced by a 137-Cs source did not change the quantity of the stable free radicals, indicating that the ionizing radiations do not generate in the molecule new free radical species [74]. Moreover, it has been proven that radiation exposure affects the redox potential of EU and causes an electron transfer that maintains the polymer in a constant oxidation state. This prolonged oxidation results in a current production and in an extended redox cycling capacity [75]. Further investigations have pointed out that EU extracted from irradiated cells acts more efficiently as electron-transfer agent, increasing the transfer velocity up to four times compared to the unexposed ones [76,77].

Whereas all these findings helped to interpret the behavior of irradiated EU, a more complete understanding of the protective functions exerted by EU has been reached by considering the occurrence of a further phenomenon, the Electron Compton Scattering (ECS), which is a process discovered by A.H. Compton in 1923 [78]. Revskaja et al. [79] suggested that the ECS process, i.e., the inelastic scattering of photons by charged particles, could act in conjunction with the other redox-based mechanisms to protect living species from radiation-induced damages.

The high degree of radical stability evidenced in EU under highly energetic radiations has led to the assumption that EU prevents such dangerous radiations from generating and spreading further radical species [80]. The most probable chemical/physical scheme is that the scattered Compton electrons dissipate energy interacting with the π-electrons rich sub-units of the EU aromatic structure, making more efficient their trapping by the stable free radicals population present in the EU backbone [81]. As a whole, the attenuation by Compton scattering of the energy released by high-energy radiations would impede the occurrence of secondary ionizations and the starting of the radical chains responsible of multiple DNA damages [82].

The extreme ability of melanin to resist radiation-induced stresses has been widely tested by exposing melanized species to different types of radiations. The highest number of experiments has been performed on the cells of the cryptoendolithic black fungus *Cryomyces antarcticus*, which has been irradiated by high doses of x- and γ-rays [83], of beta particles [81], of alpha and deuteron particles [84], of He-ions [85] and also of accelerated Fe-ions [86]. These organisms, as well as other extremophilic terrestrial species such as the melanized fungus *Friedmanniomyces endolithicus* [87], demonstrated not only a high tolerance to acute doses of radiations but also an increase in metabolic activity.

In addition to its singular radioprotective function, in fact, EU has been found able to convert radioactivity in energy available for metabolic processes, which is an ability termed radiotropism [88]. This unexpected feature has been demonstrated through a series of interesting effects, as the faster incorporation of acetate by melanized fungi under irradiation [23,89] and the upregulation, after irradiation, of ribosomal biogenesis genes in melanized yeast [90]. The large number of experiments performed on dark fungal species in the last few years have confirmed the radiation-induced phenomenon, ascribing the increased proliferation of irradiated species to the unusual mechanism of energy conversion put into play by EU [77,81,91,92,93].

What is even more interesting is that the radioprotective effect of EU, either natural or artificial, can be transferred to organisms not able to produce this biopolymer [80]. Higher survival rates after γ-irradiation have been indeed measured for non-melanized fungal cells grown in EU-containing cultures media and for mice models subjected to intravenous injection or the ingestion of EU [94,95,96].

It is noteworthy that the radioprotection conferred by EU may explain the abundance of highly melanized fungal spores found in early Cretaceous period deposits, when many other vegetal and animal classes did not survive the high radiation levels [97,98].

## 5. The Good and the Bad: Playing Conflicting Roles

The research carried out in the last few years has deeply modified the conventional picture of naturally occurring EU just as a protective agent against harmful UV radiations [99], highlighting the many fundamental roles played by this biopolymer to sustain the vital functions of almost all the terrestrial species. In those studies, the stability and the resistance of fungal biomarkers exposed to conditions simulating those of deep space and of Mars have led to speculation that EU could support also some kind of extraterrestrial life [100,101].

However, the highly beneficial abilities of natural EU that help living species to survive and also thrive under adverse conditions are also matched by contrasting and often unfavorable abilities. Under stress, EU shows indeed the feasibility to choose alternative pathways or to reverse its chemical/physical processes. The switching from a function to an opposite one results in the well-known radiative/non-radiative relaxations, hybrid electronic/protonic conductivities, metals uptake/release capabilities and radioprotective/radiotropic actions. Any of these competing behaviors influences the redox activities of the biopolymer [102] and contributes to the modulation of anti-oxidant or pro-oxidant actions [35,73]. A scheme showing the EU’s redox properties is reported in Figure 3.

Recent research studies performed by advanced methodologies highlighted the duality of roles played by EU and helped to fully understand how this pigment, in the presence of stresses, can implement protective or conversely damaging mechanisms.

It has long been known that EU is able to absorb the whole UV-visible portion of the electromagnetic spectrum and has a complex mechanism of relaxation that helps in dissipating the energy of the impinging photons [99]. However, more recently, it has been demonstrated that the UV-visible absorption coefficients are strongly dependent on the DHICA/DHI ratio [103]. The variability of this ratio, as well as the mutual organization of such building blocks in the melanosomes and the distribution of these last at the cell surface, affects the decays of the channels activated by UV-visible light. This occasionally leads to reverting the photoactive process of cell protection against oxidative stresses, inducing instead phototoxic effects [103].

A series of optical absorption and photoconductivity experiments had in the past evidenced in dry EU a wavelength-dependent switching from a positive to a negative photoconductivity [104]. This behavior and the broad spectrum of the photonegative effect suggested the existence of shallow and deep trap states near the conduction band above the Fermi level and of recombination centers below the Fermi level. More recent researchers have confirmed this model, which would explain the occurrence of transitions to the trap levels not only from the valence band but also from the localized levels above the valence band [22].

In both natural and synthetic EU, the decay of optical active states after the absorption of ultraviolet and visible radiation occurs prevalently through non-radiative dissipation, with the radiative quantum yield depending on excitation energy [105]. Femtosecond transient absorption spectroscopy used to monitor the relaxation features of EU building blocks showed that the excited singlet state of the DHI monomer decays through two main channels: fluorescence and cation radical formation. The first pathway provides a rapid deactivation of the excited state and has implications for the photoprotective role exerted by eumelanin [106]. The decay of cation radicals and of the triplet states generated by UV-A and UV-B radiations occur instead on a longer time scale [60]. This means that upon light irradiation, EU behaves transiently as a pro-oxidant rather than as an anti-oxidant. This would explain why in some cases not only is the traditional function of light screening inactive, but conversely, the cells containing EU are more susceptible to light-induced damages [107].

This delayed relaxation allows indeed the excited EU to take part in the reactions leading to the development of skin and eyes melanomas [44,108].

Even if the relationship between photoreactivity and melanoma pathogenesis is an issue that needs further investigation, what emerges is the rather intriguing role of EU in the induction of malignant melanomas of the skin and eyes [14]. Experiments have outlined two different wavelength-dependent pathways for UV-induced melanomas. In the case of UV-A, the induction of melanomas is caused by a mechanism of energy transfer from the EU aggregates (melanocytes) to DNA nucleotides. Conversely, melanomas induced by UV-B follow a pigment-independent route, with radiations provoking damages through the direct excitation of DNA nucleotides [109].

Premi et al. discovered a further unexpected negative role played by the EU pigment traditionally thought to be a protector against photo-induced cancers [110]. It had been indeed observed that the photogeneration of the DNA products responsible for mutation in skin cells, such as cyclobutane pyrimidine dimers, continues for hours after the light exposure [110]. Such a delayed effect has been explained by a chemi-excitation mechanism, in which the UV-generated reactive species excite electrons of the EU backbone, with a subsequent transfer of the harmful energy to the DNA. The acting of EU as an energy mediator able to induce delayed dangers in DNA would explain the so far elusive instances of cellular damages discovered in human tissues even in darkness.

Innovative spectroelectrochemical experiments have evidenced a correlation between free radical scavenging and redox properties, and they have demonstrated that the quenching of free radicals follows pathways that depend on the oxidation state of EU [18]. Actually, the scavenging of radicals is a task performed by the EU in a reduced state giving up electrons and by EU in an oxidized state gaining electrons [12]. The photoaging puts into action a series of reversible and nonreversible oxidation processes that induce changes in the redox equilibrium of the DHI and DHICA sub-units. It is now established that the alterations of the EU redox properties have a strong impact on various severe pathologies [21,111,112].

The redox equilibrium between the EU sub-units can be altered also by other stimuli, *in primis* by pH variations. This results in a further remarkable feature of EU: the dual electronic and protonic semiconductivity. The contribution of these mechanisms to the macroscopic conductive properties of EU mainly depends on the hydration level of the biopigment, and the switching from a hybrid to a predominant protonic conduction is regulated by the local density of the two potential charge carriers, namely the protons and the semiquinone free radicals [14].

The proton-induced shift of the comproportionation redox equilibrium between hydroquinone and quinone forms, with a boosted production of semiquinone species, results in a conductivity of hydrated EU higher by about three orders of magnitude than that of dry samples [102]. Overall, due to its peculiar redox behavior, the biopolymer is able to increase its conductivity and to switch the direction of electrons transfer, inducing in such a way anti-oxidant protective effects or conversely pro-oxidant pathogenic effects [11,18].

Further adjustments of both electrical response and radical scavenging activity are accomplished by EU also by a mechanism of metal uptake at well-defined molecular sites [19]. It is interesting to note indeed that the metallic species can interact with phenolic, aminic or carboxylic groups of the indole units, generating several different sites where metals from extracellular sources can be selectively accumulated and confined [113]. The variability of both accumulation sites and amounts of trapped metals affects the charge transport properties and the ROS quenching ability of EU, occasionally reducing its protective effect against oxidative stress [21]. While the segregation of redox-active metals by EU minimizes the feasibility to generate oxidizing species in mammalian brains, the disruption of the redox equilibrium triggered by a still not defined mechanism of metals release has been associated with neuroinflammation and neurodegeneration processes and a possible occurrence of neurologic diseases, in particular Parkinson’s [114]. In addition, some age-related eye pathologies are likely associated to altered metal ions–Eu interactions [115].

Even if protonic and mixed protonic/electronic conduction is a property rather widespread in the plant world [22] and many other biomolecules exert anti-oxidant effects, EU appears unique in its adapting the responses accordingly to the environmental conditions and in adjusting them following natural or man-made variations.

However, the feature which makes EU very different from every other organic compound is the ability not just to withstand high energy ionizing radiations but also to convert a potentially dangerous energy in a contribution to the metabolism of living species. According to current hypotheses, this task is accomplished by EU by putting in action, besides the conventional free radical quenching, also a more sophisticated physical process, i.e., the Compton scattering of electrons, that induces a gradual attenuation of primary radiation energy. Stimulated by ionizing radiations, EU would take advantage of a complex response mechanism that combines chemical, physical and biological actions to control excitation/deexcitation steps, to quench cytotoxic free radicals and to activate radio-synthesis processes [81].

The observation that melanized species exposed to ionizing radiations manage to activate a front-line radio-adaptive response by increasing their EU production has led to considering the opportunity to utilize nature-inspired melanin-based materials for human radioprotection [74,116].

The transfer of the radioprotection provided by either natural or artificial EU to not-intrinsically melanized organisms is a promising application in radiology and radiotherapy [117]. Melanin-based radiation shielding and the internal administration of EU are also planned to contrast the critical effects of cosmic rays in deep space manned flights and in future Moon and Mars missions/colonizations [118,119].

Nevertheless, the complex interplay between free radical scavenging and the Compton process, that assures a radioprotective effect, acts also by supplying energy available for metabolic processes and boosting therefore the growth of melanized species. The paramount conversion of harmful radiations in energy for life represents certainly, from a certain point of view, a positive outcome, but it has also negative sides.

Whereas it is easy to image the advantages of producing bio-fuel from fungal species and also of feeding astronauts with dark eatable mushrooms during long missions [120], less obvious are the other implications that such anomalous fungal growth have on human and animal diseases. As highlighted in the review by Revankar et al. [121], approximately 70 genera of melanized fungi behave indeed as etiologic agents, and several studies have revealed that melanized cells tend to be more aggressive than the unmelanized ones [122,123,124]. Moreover, the process of infection further increases the expression of the gene responsible for melanin production. Added to this vicious loop is the alteration of the host immune response, as EU is able to hamper the phagocytic oxidative burst and to inhibit cytokine production. As a result, in environments characterized by high levels of ionizing radiations, both the proliferation rate and spore generation of melanized pathogen yeasts, fungi and bacteria are strongly enhanced. Considering that these species act very often synergistically, their proliferation can cause severe threats to all living bodies exposed to radiations, such as the patients undergoing radiotherapy treatments [125], military personnel in nuclear environments [126] or crew members of long spatial explorations [127,128].

A further negative consequence of the radiation-enhanced growth regards the difficulty of controlling the unwanted populations of melanized harmful microorganisms via traditional x-ray sterilization techniques.

## 6. Last Considerations: Still Open Questions

It is difficult, if not impossible, to draw definitive conclusions on what EU represents in our living world. This is not a scientific/technical problem, because the adoption of advanced methodologies has enabled us to disclose the main characteristics of this unusual biopolymer, and currently, there is a general consensus on its structure and properties. Whereas a plethora of experiments provide convincing support for a photoprotective activity, many other activities exerted by EU still await to be clarified and, above all, rationalized.

What instead is still lacking is a comprehensive view of how EU, that permeates virtually all terrestrial environments, can be put into the evolutionary biological context.

From the beginning of the bio-era, adaptive processes were involved in developing the chemistry most suited to address the deep changes and the markedly different conditions experienced by the Earth. Aiming to make life compatible with the environmental conditions, evolution has continuously generated, shaped, developed and destroyed chemical structures. However, the similarity of melanin found in fossilized organisms dating to the late Carboniferous with the EU of extant organisms seems to indicate that the highly efficient activity of selection and adaptation has left this biopolymer nearly untouched. Whereas dinosaurs had extinguished, melanosomes preserved within their fossilized integumentary structures show chemical/ compositional/structural features very similar to those that drive the peculiar behavior of the today’s melanosomes

The surprising maintenance of the key features and properties for millions of years, whereas evolution modified almost all the forms of life, can likely mean that EU from the very beginning had been able to perform at best the tasks assigned by nature.

It can be assumed that the origin of the opportunities offered by EU to the melanized organisms lies in its not definite structure, which is well beyond a conventional randomly arranged polymer-like layout. The inherent disorder of EU arises indeed not only from the variable ratios between the monomers, from their cross-linking at different sites, from the varying numbers of monomers into substructures and from different stacking modes of these last, but also from the variable redox states of the constituting monomers. These “geometrical” and “electronic” variabilities result in the ability to generate countless molecular arrangements in response to external stimuli and to facilitate a variety of functions. The fact that EU already in the early biological age offered excellent solutions for viability could be a clue to explain why the adaptation processes did not significantly affect such a versatile, multitasking macromolecule. However, this for now remains an unaddressed issue.

What is well known, instead, is that EU continued by then to put into play all its sophisticated mechanisms to protect melanized species from highly dangerous stresses. In this context, the conversion of ionizing radiations in metabolic activity is the most intriguing and really unique property of EU. In effect, even if there are living species that survive under challenging conditions, only the organisms that adopt the strategy to secrete EU and accumulate the granules inside their cells are able to thrive and flourish in radiation-rich environments, such as those of our early Earth.

On the whole, it is evident that melanization has represented an important mechanism of life adaptation to Earth’s climatic and environmental changes, and this would explain also why this pigment is found so widespread in living species like no other macromolecule.

However, it is still unknown why occasionally the interactions of melanosomes with the surrounding environment cause an alteration of the redox equilibrium, reversing the protective/beneficial role of EU, which then becomes a dangerous aggressor for the host organism. While we have now a rather good knowledge of how EU, using the flexibility offered by the changeable features of its sub-units, governs its redox functionalities, still unexplained is what triggers the modification responsible for adverse, sometimes very detrimental effects. In particular, it is still unknown if this odd dual behavior of EU is due to a random chance or is driven by mechanisms of natural selection and competition for survival.

In any case, we expect to discover new aspects and to define new bio-related functionalities for such versatile, fascinating biopolymers that, likely traced back up to prebiotic times, is showing also great promise as an advanced multitasking material for our present technological world.

## Figures and Tables

**Figure 1 ijms-24-07783-f001:**
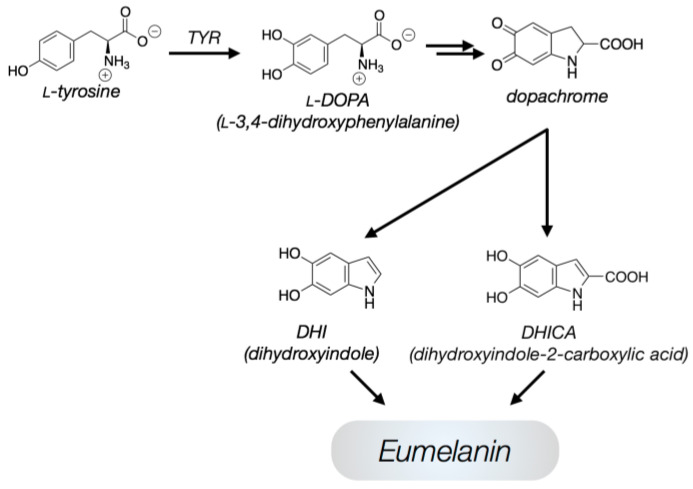
A simplified version of the synthetic pathway leading to the DHI and DHICA oligomers. (Adapted from Ref. [1]. Copyright© 2017 by the authors. Licensee MDPI, Basel, Switzerland).

**Figure 2 ijms-24-07783-f002:**
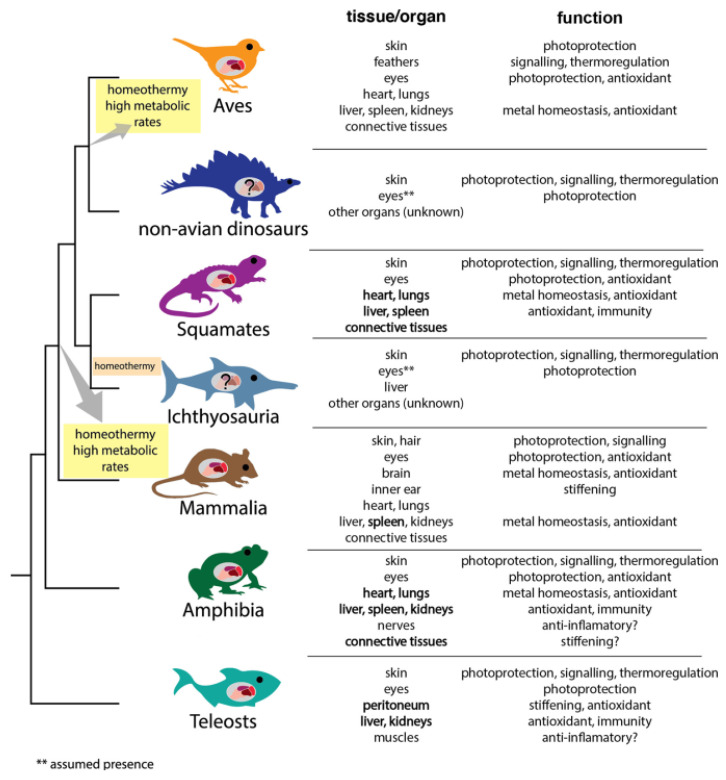
Summary of current understanding of the location of melanosomes in different body tissues and organs and putative functions in major vertebrate classes (reproduced from Ref. [53]. Copyright© 2021 The Authors. Licensee Elsevier Ltd.).

**Figure 3 ijms-24-07783-f003:**
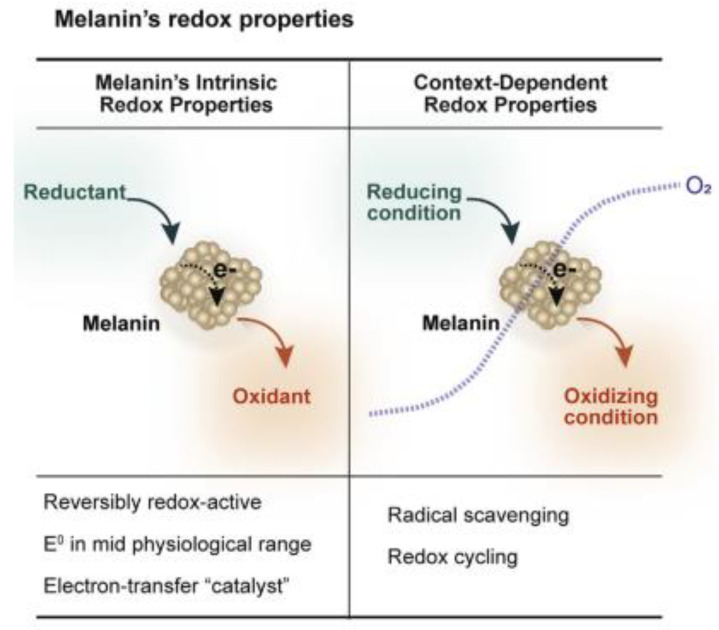
Reversibility of redox processes and radical scavenging properties of EU revealed through mediated electrochemical probing (Adapted with permission from Ref. [73]. Copyright© 2019 The Authors. Published by Elsevier Ltd.).

## Data Availability

Not applicable.

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
