# Peer review of "Prominent Roles and Conflicted Attitudes of Eumelanin in the Living World"

_ijms, 2023, doi:10.3390/ijms24097783_

Round 1
Reviewer 1 Report
I would like to suggest only to add a further comment about another reference: Sylvain Dubey & Alexander Roulin, Evolutionary and biochemical consequences of internal melanins, Pigment Cell Melanoma Res, 2014, 27(3), 327-338. https://doi.org/10.1111/pcmr.12231
In this paper the Authors emphasize the role of internal melanins not only in human beens but also in other livimg species. Moreover, they report that several sudies have shown that the amount of melanin deposited on the external body surfabe is correlated with the amount located inside the body.
pag.3 line 18. "......This odd feature is likely related...." and NOT "....This odd feature il likely related...."
Author Response
REVIEWER 1
-I would like to suggest only to add a further comment about another reference: Sylvain Dubey & Alexander Roulin, Evolutionary and biochemical consequences of internal melanins, Pigment Cell Melanoma Res, 2014, 27(3), 327-338. https://doi.org/10.1111/pcmr.12231
In this paper the Authors emphasize the role of internal melanins not only in human beens but also in other livimg species. Moreover, they report that several sudies have shown that the amount of melanin deposited on the external body surfabe is correlated with the amount located inside the body.
Following this suggestion the reference by Dubey and Roulin has been added to the text ( Ref 51), with the related comment.
-pag.3 line 18. "......This odd feature is likely related...." and NOT "....This odd feature il likely related...."
This error has been corrected in the revised version.
Reviewer 2 Report
The manuscript is undoubtedly interesting and pleasant to read for its content, since it provides new points of view and perspectives in the fascinating and still in part obscure field of melanin research. I would just suggest the author to carefully re-read the manuscript to correct some typing errors throughout the text to further improve its readability.
The following are other recommendations:
1) Figure 1: “O2” must be removed from the arrow indicating the conversion of dopachrome to DHICA (it is not an oxidation, but a simple tautomerization).
2) The following sentence on page 2 should be rephrased because it is not very clear: “Inside the biopolymer can be found also small amounts of indole structures, formed by an aromatic ring of pyrrole di-carboxylic acid and a pyrrole ring of tricarboxylic acids” (pyrroles are not indoles)
3) I would not use the term “molecular configuration” (page 3 and others), since “configuration” has a specific meaning in organic chemistry.
4) A sentence should be added at the end of Introduction to briefly present the aim of the present review.
5) I would change the title of section 4 in order to make it more consistent with the language style of the titles of the other sections, and I would remove the subdivision in the subsections 4.1 and 4.2.
6) Page 6: singlet oxygen is not a radical, please correct this.
7) Page 6: “Fe(2+)” should be “Fe2+” (without brackets).
8) The following sentence on page 7 should be rephrased because it is not very clear: “Moreover it has been proven that the radiation-induced oxidation states remain constant for a long period, causing a significant alteration of the redox potentials , producing a current and moreover extending the reversibility of the redox cycles” (it is not clear how a constant oxidation step may cause significant alterations of the redox potentials)
9) The following sentence on page 8 should be rephrased because it is not very clear: “A large number of experiments have provided an explanation for the radiation-induced extraordinary growth of some dark fungal species , linking their increased proliferation to the unparalleled functional responses of EU” (what does “unpararalleled functional responses” mean?)
10) Section 5: a figure should be added to help the reader follow the text.
11) The following sentence on page 8 should be rephrased because it is not very clear: “However , the lack in uniformity that imparts stunning abilities to natural EU gives rise also to contrasting, sometimes odd behaviours”.
Author Response
REVIEWER 2
The manuscript is undoubtedly interesting and pleasant to read for its content, since it provides new points of view and perspectives in the fascinating and still in part obscure field of melanin research. I would just suggest the author to carefully re-read the manuscript to correct some typing errors throughout the text to further improve its readability.
The author thanks the Reviewer for this positive comment and the appreciation of the manuscript .The author would like to thank this Reviewer for having attracted her attention to some points that were not adequately highlighted in the first draft and for his/her right , accurate and precise comments, that certainly will help to improve the readibility of the text .
The following are other recommendations:
- Figure 1: “O2” must be removed from the arrow indicating the conversion of dopachrome to DHICA (it is not an oxidation, but a simple tautomerization).
The Reviewer is absolutely right, I had reproduced the figure from Ref.1 without noting the error. Figure 1 has been corrected .
- The following sentence on page 2 should be rephrased because it is not very clear: “Inside the biopolymer can be found also small amounts of indole structures, formed by an aromatic ring of pyrrole di-carboxylic acid and a pyrrole ring of tricarboxylic acids” (pyrroles are not indoles)
This sentence has been rephrased .
- I would not use the term “molecular configuration” (page 3 and others), since “configuration” has a specific meaning in organic chemistry.
The Reviewer is right, the term “configuration” has been replaced by “arrangement” .
- A sentence should be added at the end of Introduction to briefly present the aim of the present review.
This also is a good suggestion, a short sentence has been added at the end of the introduction section.
- I would change the title of section 4 in order to make it more consistent with the language style of the titles of the other sections, and I would remove the subdivision in the subsections 4.1 and 4.2.
Following this suggestion the title of Section 4 has been changed and the subsections have been removed.
- Page 6: singlet oxygen is not a radical, please correct this.
Right comment , the term “radical” has been replaced by “reactive species”
- Page 6: “Fe(2+)” should be “Fe2+” (without brackets).
Made !!!
- The following sentence on page 7 should be rephrased because it is not very clear: “Moreover it has been proven that the radiation-induced oxidation states remain constant for a long period, causing a significant alteration of the redox potentials , producing a current and moreover extending the reversibility of the redox cycles” (it is not clear how a constant oxidation step may cause significant alterations of the redox potentials)
I apologize, in effect this sentence was rather messy .The whole sentence has been rephrased in the revised version.
- The following sentence on page 8 should be rephrased because it is not very clear: “A large number of experiments have provided an explanation for the radiation-induced extraordinary growth of some dark fungal species , linking their increased proliferation to the unparalleled functional responses of EU” (what does “unpararalleled functional responses” mean?)
The same as above, the sentence has been rephrased.
- Section 5: a figure should be added to help the reader follow the text.
To clarify this important point raised by the Reviewer a new figure has been added (Fig.3).
- The following sentence on page 8 should be rephrased because it is not very clear: “However , the lack in uniformity that imparts stunning abilities to natural EU gives rise also to contrasting, sometimes odd behaviours”.
The same as points 9) and 10) , the sentence has been rephrased.